# Comparison of Hepatic Tissue Characterization between T1-Mapping and Non-Contrast Computed Tomography

**DOI:** 10.3390/jcm11102863

**Published:** 2022-05-19

**Authors:** Constanze Bardach, Leonie Morski, Katharina Mascherbauer, Carolina Donà, Matthias Koschutnik, Kseniya Halavina, Christian Nitsche, Dietrich Beitzke, Christian Loewe, Elisabeth Waldmann, Michael Trauner, Julia Mascherbauer, Christian Hengstenberg, Andreas Kammerlander

**Affiliations:** 1Division of Cardiovascular and Interventional Radiology, Medical University of Vienna, 1090 Vienna, Austria; constanze.bardach@meduniwien.ac.at (C.B.); dietrich.beitzke@meduniwien.ac.at (D.B.); christian.loewe@meduniwien.ac.at (C.L.); 2Division of Cardiology, Medical University of Vienna, 1090 Vienna, Austria; n11817681@students.meduniwien.ac.at (L.M.); katharina.mascherbauer@meduniwien.ac.at (K.M.); carolina.dona@meduniwien.ac.at (C.D.); matthias.koschutnik@meduniwien.ac.at (M.K.); kseniya.halavina@meduniwien.ac.at (K.H.); christian.nitsche@meduniwien.ac.at (C.N.); julia.mascherbauer@meduniwien.ac.at (J.M.); christian.hengstenberg@meduniwien.ac.at (C.H.); 3Division of Gastroenterology and Hepatology, Medical University of Vienna, 1090 Vienna, Austria; elisabeth.waldmann@meduniwien.ac.at (E.W.); michael.trauner@meduniwien.ac.at (M.T.); 4Department of Internal Medicine 3, University Hospital St. Pölten, Karl Landsteiner University of Health Sciences, 3500 Krems, Austria

**Keywords:** MAFLD, NAFLD, computed tomography, cardiovascular magnetic resonance imaging

## Abstract

Background: Non-contrast computed tomography (CT) is frequently used to assess non-alcoholic/metabolic fatty liver disease (NAFLD/MAFLD), which is associated with cardiovascular risk. Although liver biopsy is considered the gold standard for diagnosis, standardized scores and non-contrast computed tomography (CT) are used instead. On standard cardiac T1-maps on cardiovascular imaging (CMR) exams for myocardial tissue characterization hepatic tissue is also visible. We hypothesized that there is a significant correlation between hepatic tissue T1-times on CMR and Hounsfield units (HU) on non-contrast CT. Methods: We retrospectively identified patients undergoing a non-contrast CT including the abdomen, a CMR including T1-mapping, and laboratory assessment within 30 days. Patients with storage diseases were excluded. Results: We identified 271 patients (62 ± 15 y/o, 49% female) undergoing non-contrast CT and CMR T1-mapping within 30 days. Mean hepatic HU values were 54 ± 11 on CT and native T1-times were 598 ± 102 ms on CMR and there was a weak, but significant, correlation between these parameters (r = −0.136, *p* = 0.025). On age and sex adjusted regression analysis, lower liver HU values indicated a dismal cardiometabolic risk profile, including higher HbA1C (*p* = 0.005) and higher body mass index (*p* < 0.001). In contrast, native hepatic T1-times yielded a more pronounced cardiac risk profile, including impaired systolic function (*p* = 0.045) and higher NT-proBNP values (N-Terminal Brain Natriuretic Peptide) (*p* = 0.004). Conclusions: Hepatic T1-times are easy to assess on standard T1-maps on CMR but only weakly correlated with hepatic HU values on CT and clinical NAFLD/MAFLD scores. Liver T1-times, however, are linked to impaired systolic function and higher natriuretic peptide levels. The prognostic value and clinical usefulness of hepatic T1-times in CMR cohorts warrants further research.

## 1. Introduction

Metabolic dysfunction associated, or fatty, liver disease (MAFLD, NAFLD) describes a continuum of hepatic diseases ranging from steatosis to liver cirrhosis. MAFLD/NAFLD is the most prevalent liver disease, affecting one in four individuals in a Western population. So far, no treatment is available [1]. MAFLD/NAFLD is well known to be associated with cardiovascular (CVD) disease, [2] sharing several cardiometabolic risk factors [3].

Despite its high prevalence and clinical importance, the diagnosis of MAFLD/NAFLD is challenging. Liver biopsy represents the gold standard for MAFLD/NAFLD diagnosis but is infrequently performed due to its invasive nature. Alternatively, the NAFLD Fibrosis score (NFS) [4] and Fibrosis-4 Index [5] incorporate several clinical and laboratory parameters allowing to estimate the liver fibrosis stage. An NFS of greater than 0.675 indicates severe fibrosis or cirrhosis and a Fib-4 Score greater than or equal to 2.67 indicates advanced fibrosis.

Moreover, non-invasive imaging is frequently used to estimate the risk for MAFLD/NAFLD. Non-contrast computed tomography (CT) and magnetic resonance imaging (MR) aim to detect hepatic tissue signal and liver stiffness alterations due to increased fibrosis [2,6]. CT is commonly used to estimate the risk for MAFLD/NAFD. The attenuation value of liver parenchyma measured in Hounsfield units (HU) is decreased by the presence of hepatic steatosis. A normal liver attenuation is approximately 60 HU, with attenuation values below 40 HU, or a liver-to-spleen HU ratio of <1.0, often used to define MAFLD/NAFLD [7].

Several factors have been proposed to promote MAFLD/NAFLD, many of which play an important role in CVD, including endothelial dysfunction, altered lipid metabolism, systemic inflammation, oxidative stress, and systemic insulin resistance [8,9].

In patients with CVD, cardiovascular magnetic resonance imaging (CMR) is evolving as the first-line imaging technique as it is the gold standard for structural and functional assessment of the heart. In addition, cardiac tissue characterization by T1-mapping provides novel insights into pathological alterations within the myocardium [10,11]. Thus far, only few studies report on the use of T1-mapping in liver MR scans but show promising results [12,13,14]. On standard cardiac T1-maps (Figure 1), liver tissue is also assessable but so far has not been used to study hepatic T1-times.

Preclinical data [15] and one small study in patients with congenital heart disease suggest that liver T1-mapping may be feasible and useful in cardiac disease [16]. However, hepatic T1-mapping on standard CMR studies has not yet been performed in a well-described cardiology cohort.

## 2. Materials and Methods

### 2.1. Study Design

This was a retrospective study in a single tertiary referral center. Using our hospital-wide electronic system, we identified individuals who had undergone within 30 days: (1) non-contrast CT scans including the abdomen allowing for hepatic tissue assessment, and (2) CMR including T1-mapping where hepatic tissue is visualized in a standard mid-cavity short-axis slice (Figure 1). The institutional review board (EK Nr. 2039/2020) approved the study protocol. Informed consent was waived due to the retrospective nature.

### 2.2. Clinical Definitions

At the time of CMR and CT, demographic data (age, sex, body mass index, body surface area) and comorbidities were assessed. These included hypertension (≥140/90 mm Hg or antihypertensive treatment), atrial fibrillation (present at the time of CMR or documented in the medical history), diabetes (fasting blood glucose level >126 mg/dL, HbA1c over 6.5, or use of anti-diabetic medication), hyperlipidemia (total serum cholesterol 240 mg/dL or use of cholesterol-lowering medication), smoking status, coronary artery disease (CAD), previous percutaneous coronary intervention, and previous coronary artery bypass grafting (CABG). Previous myocardial infarction was defined both by medical history and CMR. The estimated glomerular filtration rate was calculated with the simplified Modification of Diet in Renal Disease formula [17]. In addition, we assessed laboratory values, including NT-proBNP, albumin, creatinine, platelet count, prothrombin time (PT), alkaline phosphatase (AP), aspartate aminotransferase (AST), alanine aminotransferase (ALT), gamma-glutamyl transferase (gGT), international normalized ratio (INR), and sodium levels, if available within 30 days of the CMR and CT scans.

### 2.3. Cardiovascular Magnetic Resonance Imaging

All patients underwent CMR examinations on a 1.5-T scanner (MAGNETOM Avantofit; Siemens Healthineers GmbH, Erlangen, Germany) following standard protocols that included late gadolinium enhancement (LGE) imaging [0.15 mmol/kg gadobutrol (Gadovist; Bayer Vital GmbH, Leverkusen, Germany)] if the estimated glomerular filtration rate was >30 mL/min/1.73 m^2^.

T1 mapping was performed with electrocardiographically triggered MOLLI (Modified Look-Locker Inversion Recovery) using a 5(3)3 prototype (5 acquisition heartbeats followed by 3 recovery heartbeats and a further 3 acquisition heartbeats) on a short-axis mid cavity slice and with a 4-chamber view. This method generates an inline, pixel-based T1 map by acquiring a series of images over several heartbeats with shifted T1 times, inline motion correction, and inline calculation of the T1 relaxation curve within one breath. T1-sequence parameters were as follows: starting inversion time (TI) 120 ms, TI increment 80 ms, reconstructed matrix size 256 × 218, measured matrix size 256 × 144 (phase-encoding resolution 66%, phase-encoding field of view 85%). T1 maps were created both before and 15 min after contrast agent application.

Hepatic T1-times were measured using an average of three measures with sufficient distance to vessels and the bile duct (Figure 1). Myocardial T1 times were averaged from a mid-cavity short-axis and a 4-chamber view with sufficient distance to the blood pool.

### 2.4. Computed Tomography

CT scans were performed on 64-MDCT (Multi-Detector Computed Tomography) scanners at 100–120 kV with automatic tube current modulation and 5 × 3 mm reconstructions (Somatom Force/Drive; Siemens Healthineers, Erlangen, Germany). We only used the unenhanced images for assessment of liver attenuation. We defined three regions of interest on an axial single slice, two placed in the right liver lobe and one in the left liver lobe. The regions of interest had a maximum diameter of approximately 1 cm and regions close to vessels or the bile duct were avoided. The distance to the Glisson’s capsule was a minimum of 2 cm. We used an average of these three measurements.

### 2.5. NAFLD Fibrosis Score

The NFS was calculated with its formula: NFS = −1.675 + (0.037 × age [years]) + (0.094 × BMI (Body Mass Index) b [kg/m^2^]) + (1.13 × IFG/diabetes [yes = 1, no = 0]) + (0.99 × AST/ALT ratio) – (0.013 × platelet count [×10^9^/L]) – (0.66 × albumin [g/dL]). Results were stratified by the cut-off values < −1.455 (mild to moderate fibrosis), −1.455 to 0.675 (indeterminant score), and a score of >0.675 (severe fibrosis or cirrhosis) [18].

### 2.6. Statistical Analysis

All statistical analyses were computed using Stata 15. Continuous data are expressed as mean ± standard deviation or median with corresponding interquartile range, and categorical variables are presented as percentages or total numbers. We used intraclass correlation coefficients (ICC) to assess inter-rater variability, for which a second, blinded reader performed measurments on 50 randomly chosen patients.

Pearsons’s correlation coefficients were used to assess an association between liver attenuation on CT and T1-mapping. The association between native hepatic T1-times and established markers of cardiac risk, including left and right ventricular ejection fractions (LVEF, RVEF) and NT-proBNP was tested using age and sex adjusted regression analysis.

For all analyses, the level of significance was set to 0.05.

## 3. Results

### 3.1. Demographics

We identified 1045 individuals undergoing native CT and CMR, of whom 135 were excluded due to inadequate imaging quality. Out of the remaining 910 patients, 271 individuals underwent CT and CMR within 30 days, forming our final study cohort. Indications for CMR were heart failure (*n* = 77, 28%), left ventricular hypertrophy (*n* = 77, 28%), assessment of valvular heart disease (*n* = 45; 17%), coronary artery disease (*n* = 22, 8%), myocarditis (*n* = 15, 6%), and others (*n* = 35, 13%).

### 3.2. Association between Liver Tissue Characteristics on CT and CMR

Mean HU values of the liver on unenhanced CT scans were 55 ± 11 HU and mean native T1-times of the liver on CMR were 590 ± 96 ms. Inter-rater variability was very good for both CT and CMR characterization of hepatic tissue (ICC for HU on CT: 0.87 ICC for native T1-times on CMR: 0.86). Overall, we found a weak but significant correlation between hepatic tissue characterization on CMR and CT (T1 in ms and HU: r = −0.13, *p* = 0.03) (see Figure 2)

### 3.3. Association between Hepatic Tissue Characteristics with Baseline Variables

Table 1 displays patients stratified by medians of liver HU values on CT (49 HU) and native hepatic T1 times on CMR (588 ms). Overall, baseline characteristics were well balanced across groups, however, in patients with hepatic HU values below or equal to the median, we observed a significantly higher rate of diabetes (31% versus 15%, *p* = 0.003) and a higher NFS (*p* = 0.008) when compared to patients with HU values above the median. This was not observed for hepatic T1-times on CMR. However, in contrast to CT HU values, patients with T1-times above the median were more often in atrial fibrillation (43% versus 25%, *p* = 0.003) and had higher gGT values (77 ± 93 versus 44 ± 45, *p* < 0.001).

In age- and sex-adjusted linear regression models (see Table 2), we found a significant association between hepatic T1-times and total bilirubin (*p* = 0.025), AST/GOT (Glutamat-Oxalacetat-Transaminase) (*p* = 0.011), gGT (*p* = 0.002), and NT-proBNP (*p* = 0.004), LVEF (*p* = 0.045), RV (Right Ventricle) size and function (RVEF, *p* = 0.045, RVEDV/BSA (Right Ventricle End Diastolic Volume/Body Surface Area), *p* = 0.005), LA size (*p* < 0.001), and myocardial native T1-times (*p* < 0.001).

Interestingly, regression models showed a different pattern of association with baseline variables when comparing native HU values on CT and CMR. For CT, regression models revealed an association with BMI (*p* < 0.001), HbA1C (*p* = 0.005) and the NFS (*p* = 0.014), none of which were associated with native hepatic T1-times. None of the conventional CMR parameters, including LV and RV size and function were associated with liver HU values on CT.

## 4. Discussion

We report three main findings in this exploratory study of patients undergoing non-contrast CT and CMR within 30 days: (1) liver HU values on CT and native hepatic T1-times on CMR only correlate weakly with each other, (2) liver HU values on CT are associated with cardiometabolic traits, such as diabetes and the NFS, and (3) in contrast, hepatic T1-times on CMR are associated with impaired cardiac function, including LVEF, RVEF, and natriuretic peptide levels.

In patients with CVD, liver damage is frequently observed but is rarely assessed. Two clinical scenarios are most commonly discussed as causes for liver damage in these patients: (1) congestion due to cardiac backward failure, and (2) MAFLD/NAFLD, which is the most rapidly increasing cause of liver-related mortality worldwide and is strongly associated with risk for cardiovascular disease [19].

Cardiac backward failure as a cause for liver damage can easily be identified by cardiovascular imaging using CMR, allowing for assessment of left and right ventricular function, and surrogates of filling pressures, such as left atrial size. Moreover, T1-mapping of the left ventricle on CMR further enhances our understanding of cardiac dysfunction caused by diffuse alterations within the myocardium [10], which may result in liver damage. The complex interplay between cardiac and liver dysfunction, however, is incompletely understood.

In addition to heart failure as a cause for liver damage, MAFDL/NAFLD is highly prevalent in patients with cardiovascular diseases [19]. The diagnosis of MAFLD/NAFLD, however, is challenging as it reflects a broad spectrum of liver disease, ranging from steatosis, in which lipid accumulation in hepatocytes is the predominant histological characteristic, to nonalcoholic steatohepatitis (NASH), which is characterized by additional hepatic inflammation with or without fibrosis, up to liver cirrhosis. Laboratory-based scores lack accuracy [20,21] and liver biopsy, representing the gold-standard technique, is rarely performed due to its invasive nature. Instead, non-invasive imaging, including non-contrast CT, is frequently used to diagnose MAFLD/NAFLD. Fat accumulation causes reduced attenuation on non-contrast CT scans of the liver [22]. Several approaches are used, including use of HU values of hepatic tissue alone and in relation to spleen parenchyma [23,24]. Previous studies highlighted the clinical relevance of MAFDL/NAFLD and CVD. Among 5121 asymptomatic individuals undergoing coronary CT angiography, Lee et al. [25] demonstrated that MAFLD/NAFLD was associated with non-calcified coronary plaques and sudden and unexpected cardiac events. Another study in 308 healthy subjects demonstrated that participants with NAFLD exhibited echocardiographic signs of cardiac remodeling. In addition, regarding 18-Fluorodeoxyglucose positron emission tomography (^18^FDG-PET) image analysis, cardiac glucose uptake was significantly decreased in participants with MAFLD/NAFLD [26]. Taken together, MAFLD/NAFLD is associated with CVD, and both share similar risk factors [9]. However, it is unknown how MAFLD/NAFLD is best assessed in patients with established CVD and whether it is an independent harbinger of increased risk [27].

Both CT and CMR are routine imaging techniques in patients with CVD. T1-mapping is now routinely applied in patients undergoing CMR and would also allow for liver tissue assessment as it is frequently displayed on standard cardiac T1-maps. However, hepatic T1-mapping in patients undergoing CMR has never been correlated with HU values on CT. Few studies report on T1-times of the liver in patients with CVD. In line with our findings, that native hepatic T1-times are associated with LV systolic dysfunction in a study by Isaak and Praktiknjo et al. on 60 participants receiving a liver MR and a CMR. They demonstrated that participants with greater severity of liver disease also displayed more severe systolic dysfunction, and increased myocardial T1-times [13]. Similarly, higher T1-times of the liver were associated with impaired LV function in patients with non-ischemic cardiomyopathy [14].

We could show that results from non-contrast CT and T1-mapping on CMR may represent different aspects of liver damage in patients with CVD. Whereas CT detects a cardiometabolic trait with higher prevalence of diabetes and higher NAFLD score (lower HU values indicating higher risk), higher native hepatic T1-times are associated with a dismal cardiac risk profile, including impaired left and right ventricular ejection fraction, enlarged left atrial size, and higher natriuretic peptide levels. Future research is warranted to investigate the underlying processes detected on hepatic T1-mapping and whether it may be useful for improved risk prediction in patients with established CVD.

### Limitations

Several limitations merit comment. Firstly, the single center study and the retrospective nature may result in a biased cohort and results may not be generalized to other cohorts. Secondly, T1-times reflect composite alterations within the tissue and T1-mapping results do not allow for a direct conclusion about pathophysiological processes. Of note, in NAFLD/MAFLD both fat accumulation and fibrosis play a crucial role, however, fat lowers native hepatic T1-times whereas fibrosis leads to an increase in T1-times. This may explain the weak correlation between CT and CMR tissue characterization findings. Hence, changes in T1-time in the liver may be a result of several components, which could only be confirmed by invasive biopsy.

## 5. Conclusions

Hepatic T1-times are easy to assess on standard T1-maps on CMR but only weakly correlated with hepatic HU values on CT and clinical NAFLD/MAFLD scores. Hepatic T1-times, however, are linked to impaired systolic function and higher natriuretic peptide levels. The prognostic value and clinical usefulness of hepatic T1-times in CMR cohorts warrant further research.

## Figures and Tables

**Figure 1 jcm-11-02863-f001:**
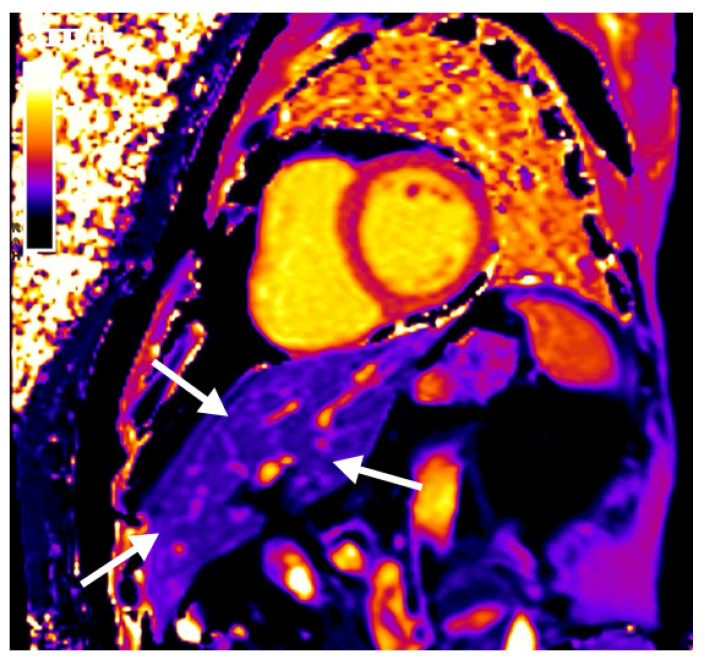
Standard short-axis T1-map on a cardiovascular magnetic resonance imaging study, allowing for assessment of hepatic tissue (white arrows).

**Figure 2 jcm-11-02863-f002:**
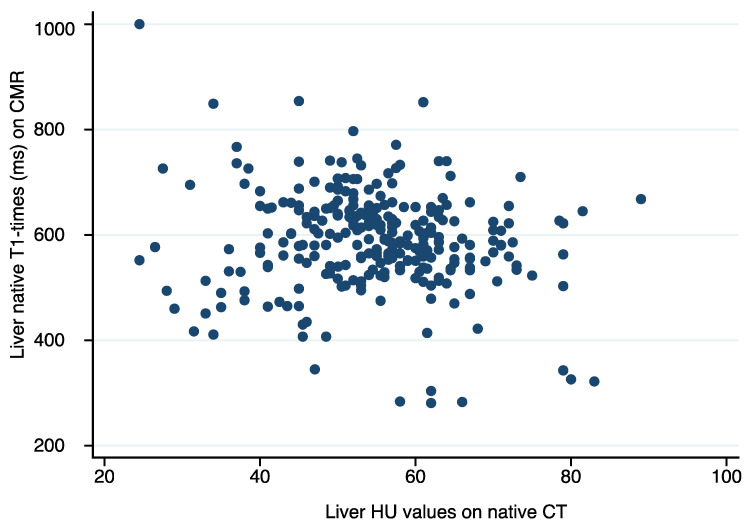
Scatter plot demonstrating the association between HU values on unenhanced computed tomography (CT) and native T1-times on cardiovascular magnetic resonance imaging (CMR) of hepatic tissue.

**Table 1 jcm-11-02863-t001:** Baseline characteristics, stratified by medians of liver HU values on computed tomography and native T1-times on cardiovascular magnetic resonance imaging.

	HU Values on CT			Native Liver T1 Times on CMR		
	<Median	>Median	*p*	<Median	>Median	*p*-Value
	*n* = 140	*n* = 131		*n* = 137	*n* = 134	
Age (years)	62.9 (14.4)	61.1 (16.0)	0.33	62.1 (14.7)	61.9 (15.7)	0.91
Height (cm)	171 (10)	171 (9)	0.93	172 (10)	169 (9)	**0.013**
Weight (kg)	84 (18)	75 (16)	**<0.001**	83 (19)	76 (17)	**0.002**
BSA (m^2^)	1.98 (0.26)	1.87 (0.23)	**0.001**	1.97 (0.25)	1.87 (0.23)	**0.001**
BMI (kg/m^2^)	28.7 (5.8)	25.6 (4.7)	**<0.001**	27.9 (5.6)	26.5 (5.3)	**0.044**
Male sex	54%	47%	0.25	58%	43%	**0.013**
CMR Indication			0.71			0.50
CAD	7%	9%		9%	7%	
VHD	19%	14%		15%	18%	
Heart Failure	29%	27%		28%	28%	
Storage Disease	29%	28%		25%	32%	
Inflammation	5%	6%		6%	5%	
Other	11%	15%		16%	10%	
Hypertension	71%	64%	0.19	71%	64%	0.22
Atrial Fibrillation	38%	30%	0.18	25%	43%	**0.003**
Diabetes	31%	15%	**0.003**	20%	26%	0.29
Hyperlipidemia	42%	34%	0.19	39%	38%	0.92
Smoking	17%	12%	0.35	13%	16%	0.54
CAD	27%	27%	0.90	27%	26%	0.86
Previous PCI	11%	10%	0.81	10%	10%	1.00
Previous CABG	6%	6%	0.88	4%	8%	0.18
Previous Stroke	4%	5%	0.70	4%	5%	0.74
Previous MI	9%	12%	0.56	11%	10%	0.68
Hematocrit (%)	38.9 (5.8)	38.9 (5.6)	0.97	39.7 (6.0)	38.0 (5.2)	**0.014**
Platelet Count	232 (74)	235 (86)	0.79	233 (85)	234 (75)	0.90
PT	90 (31)	88 (27)	0.81	92 (28)	86 (32)	0.45
INR	1.36 (0.77)	1.18 (0.45)	**0.047**	1.19 (0.60)	1.35 (0.68)	0.066
Creatinine (mg/dL)	1.44 (1.41)	1.31 (1.52)	0.50	1.41 (1.45)	1.34 (1.48)	0.68
Total bilirubin	0.60 (0.47)	0.60 (0.34)	0.96	0.55 (0.27)	0.65 (0.51)	0.050
Albumin	39 (6)	40 (5)	0.077	40 (5)	39 (6)	0.067
AP	85 (49)	75 (31)	0.058	72 (26)	88 (52)	**0.002**
AST	30 (21)	33 (51)	0.58	27 (14)	36 (53)	0.059
ALT	27 (19)	32 (55)	0.34	26 (17)	33 (55)	0.19
GGT	67 (68)	54 (81)	0.16	44 (45)	77 (93)	**<0.001**
NT-proBNP (pg/mL)	2344 (6338)	1601 (3979)	0.33	1403 (3904)	2560 (6394)	0.13
HbA1c (%)	6.0 (1.0)	5.7 (0.6)	**0.035**	5.9 (0.9)	5.9 (0.8)	0.63
LA volume (mL)	117 (59)	106 (54)	0.12	104 (51)	119 (62)	**0.028**
RA volume (mL)	99 (57)	91 (48)	0.26	90 (54)	100 (52)	0.12
LVEF (%)	58 (14)	58 (12)	0.95	59 (13)	57 (13)	0.12
RVEF (%)	54 (10)	54 (9)	0.87	55 (10)	52 (10)	**0.016**
LVEDVi (mL/m^2^)	80 (28)	81 (26)	0.76	80 (26)	81 (28)	0.61
RVEDVi (mL/m^2^)	79 (26)	77 (20)	0.50	75 (19)	81 (26)	**0.045**
Liver HU on CT	46 (8)	63 (9)	**<0.001**	53 (12)	55 (12)	0.43
Native liver T1 time (ms)	603 (98)	575 (92)	**0.015**	520 (68)	661 (61)	**<0.001**
Cardiac T1 time (ms)	1008 (47)	1005 (47)	0.61	992 (46)	1021 (43)	**<0.001**
NAFLD Fibrosis Score (metric)	−0.18 (1.76)	−0.80 (1.76)	**0.008**	−0.41 (1.86)	−0.54 (1.72)	0.58
NAFLD Fibrosis Score			**0.029**			0.69
F0–F2	18%	33%		25%	25%	
Indeterminant score	57%	49%		51%	55%	
F3–F4	25%	18%		24%	19%	

HU denotes Hounsfield Unit, CT: computed tomography, CMR: cardiovascular magnetic resonance imaging, BSA: body surface area, BMI: body mass index, CAD: coronary artery disease, VHD: valvular heart disease, PCI: percutaneous coronary intervention, CABG: coronary artery bypass graft, MI: myocardial infarction, PT: prothrombin time, INR: international normalized ratio, AP: alkaline phosphatase, AST: aspartate aminotransferase, ALT: alanine aminotransferase, GGT: gamma-glutamyl transferase, LA: left atrial, RA: right atrial, LV: left ventricular, EF: ejection fraction; EDV/BSA: end-diastolic volume indexed to body surface area, RV: right ventricular, NAFLD: non-alcoholic fatty liver disease. Bold *p*-values indicate statistical significance (*p* < 0.05).

**Table 2 jcm-11-02863-t002:** Linear regression models, adjusted for age and sex, demonstrating the association of liver HU values on non-contrast computed tomography and native liver T1-times on cardiovascular magnetic resonance imaging with baseline variables.

	HU on CT				Native T1-Time on CMR			
	Adj. Beta	LCI	UCI	*p*-Value	Adj. Beta	LCI	UCI	*p*-Value
BMI	−0.173	−0.230	−0.117	**<0.001**	−0.006	−0.013	0.001	0.121
PT	−0.064	−0.684	0.556	0.836	−0.016	−0.095	0.062	0.682
INR	−0.007	−0.015	0.001	0.101	0.001	0.000	0.002	0.136
Creatinine	−0.002	−0.018	0.014	0.804	0.000	−0.002	0.002	0.700
Total Bilirubin	−0.001	−0.005	0.004	0.692	0.001	0.000	0.001	**0.025**
Albumin	0.030	−0.035	0.094	0.368	−0.005	−0.013	0.002	0.161
AP	−0.234	−0.693	0.226	0.317	0.069	0.016	0.123	**0.011**
AST	0.029	−0.395	0.453	0.892	0.040	−0.011	0.090	0.123
ALT	0.027	−0.413	0.466	0.905	0.033	−0.019	0.084	0.214
GGT	−0.397	−1.222	0.428	0.344	0.149	0.053	0.245	**0.002**
HbA1C	−0.018	−0.030	−0.005	**0.005**	−0.001	−0.002	0.001	0.329
NT-proBNP	−0.010	−0.028	0.007	0.227	0.003	0.001	0.005	**0.004**
LVEF	−0.011	−0.151	0.129	0.878	−0.017	−0.033	0.000	**0.045**
LVEDV/BSA	−0.029	−0.318	0.260	0.843	0.028	−0.006	0.062	0.105
RVEF	0.033	−0.075	0.141	0.551	−0.013	−0.026	0.000	**0.045**
RVEDV/BSA	−0.070	−0.317	0.178	0.581	0.042	0.013	0.070	**0.005**
LA volume	−0.459	−1.061	0.143	0.135	0.131	0.061	0.201	**<0.001**
RA volume	−0.404	−0.971	0.163	0.162	0.037	−0.030	0.105	0.278
Myocardial T1-time	0.277	−0.236	0.790	0.289	0.160	0.103	0.218	**<0.001**
NAFLD Fibrosis score	−0.022	−0.040	−0.005	**0.014**	−0.001	−0.003	0.001	0.375

LCI denotes lower confidence interval, UCI: upper confidence interval, HU: Hounsfield Unit, CT: computed tomography, CMR: cardiovascular magnetic resonance imaging, BMI: body mass index, PT: prothrombin time, INR: international normalized ratio, AP: alkaline phosphatase, AST: aspartate aminotransferase, ALT: alanine aminotransferase, GGT: gamma-glutamyl transferase, LV: left ventricular, EF: ejection fraction; EDV/BSA: end-diastolic volume indexed to body surface area, RV: right ventricular, LA: left atrial, RA: right atrial, NAFLD: non-alcoholic fatty liver disease. Bold *p*-values indicate statistical significance (*p* < 0.05).

## Data Availability

Data may be shared upon reasonable request to the corresponding author.

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
