# Peer review of "Comparison of Hepatic Tissue Characterization between T1-Mapping and Non-Contrast Computed Tomography"

_jcm, 2022, doi:10.3390/jcm11102863_

Round 1
Reviewer 1 Report
In this retrospective study, authors investigated the association between characterization of liver parenchyma at CT scan and MRI with T1-mapping.
Overall, the article is well organized and structured, and results are clearly described in tables and figures.
- Introduction – line 1: correct acronym FAFLD in NAFLD
- Methods – CMR subsection: better explanation of the standard MRI protocol (it could be also included in a table)
- Methods – Statistical analysis: absence of the cut-off for significant p-value
- Did you include patients with storage disease? There is a contradiction between abstract and results
- Table 1/2: bold type or asterisk to highlight significant p-values
- Please add conclusions after limitations
Author Response
Point by point response
We adapted the manuscript according to the specific formatting comments and the reviewers’ comments as listed below.
Reviewer 1.
In this retrospective study, authors investigated the association between characterization of liver parenchyma at CT scan and MRI with T1-mapping.
Overall, the article is well organized and structured, and results are clearly described in tables and figures.
R1-1 Introduction – line 1: correct acronym FAFLD in NAFLD
Answer and Action: We changed FADLD to NAFLD.
R1-2 Methods – Statistical analysis: absence of the cut-off for significant p-value
Answer and Acation: We added the following to the Methods section:
“For all analyses, the level of significance was set to 0.05. “
R1-3 Did you include patients with storage disease? There is a contradiction between abstract and results
Answer: We excluded patients with storage disease. We clarified the list of indications (changed suspected storage disease with left ventricular hypertrophy).
Action:
Results:
‘Indications for CMR were heart failure (n=77, 28%), left ventricular hypertrophy (n=77, 28%), assessment of valvular heart disease (n=45; 17%), coronary artery disease (n=22, 8%), myocarditis (n=15, 6%), and others (n=35, 13%).’
R1-4 Table 1/2: bold type or asterisk to highlight significant p-values
Answer and Action.: We highlighted significant p-values bold.
R1-5 Please add conclusions after limitations
Answer and Action: We put the Conclusions after the Limitations in the revised manuscript.
Reviewer 2 Report
The research compares
Overall the retrospective research is clinically important. Metabolic dysfunction-associated fatty liver disease (MAFLD) and non-alcoholic fatty liver disease (NAFLD) are prevalent in western society. This study compares T1 and non-contrast CT for hepatic tissue characterization. Clinicians care about when would fatty liver become liver fibrosis.
adolescence
Major points:
1. The presentation, and the English of this paper can be improved. In order to rewrite the paper, please consult a native English speaking writer. A majority of the background information in Discussion can be moved to Introduction.
2. T1 is only weakly correlated to non-contrast CT. Please interpret the result. I suspect T1 is caused by many factors, so is non-contrast CT.
3. Please elaborate how T1 timing could differentiate fibrosis from fatty liver. Is magnetic resonance spectroscopy (MRS) relevant in management of MAFLD/NAFLD?
Minor points:
1. Abstract must be improved.
Background: The main hypothesis / purpose of the research are not clear. T1 timing for hepatic tissue characterization has never been performed, but what is the potential benefits of the research? Perhaps the author think that T1 timing could avoid the ionizing radiation of CT, or perhaps the T1 timing could detect fatty liver and cardiac disease in a single scan?
Method: The first sentence in Result "We identified 271 patients ... within 30 days." is part of Method
Discussion: It is not clear if results support the hypothesis of the study?
2. Please use consistent terms. It is not clear whether the liver T1 times, the hepatic T1 times, T1 maps and T1-mapping are the same?
Use the same term thoroughout the paper.
3. Background information in Discussion can be moved to Introduction.
4. The main conclusion must be highlighted in accordance with the main hypothesis
5. Clearly define Metabolic dysfunction-associated fatty liver disease (MAFLD) and non-alcoholic fatty liver disease (NAFLD) in Introduction.
6. There are many typos in the article.
7. REFERENCE LIST is Reference
8. Use MDPI template. See https://www.mdpi.com/journal/jcm/instructions
Author Response
Point by Point Response
Reviewer 2
The research compares adolescence
Overall the retrospective research is clinically important. Metabolic dysfunction-associated fatty liver disease (MAFLD) and non-alcoholic fatty liver disease (NAFLD) are prevalent in western society. This study compares T1 and non-contrast CT for hepatic tissue characterization. Clinicians care about when would fatty liver become liver fibrosis.
R2-1 The presentation, and the English of this paper can be improved. In order to rewrite the paper, please consult a native English speaking writer. A majority of the background information in Discussion can be moved to Introduction.
Answer: We greatly appreciate this reviewer’s comment. We revised the manuscript with our two native English speaking co-authors to improve the overall quality.
Action: We made changes throughout the manuscript. To increase the visibility of substantial changes and according to the last reviewer’s comment (use of mdpi emplate), only substantial changes could be highlighted in the change track version.
R2-2 T1 is only weakly correlated to non-contrast CT. Please interpret the result. I suspect T1 is caused by many factors, so is non-contrast CT.
Answer: We appreciate this important comment. Indeed, there was a surprisingly weak correlation between HU values on non-contrast CT and T1-times on CMR. We highlighted that T1-times are influenced by several factors and that no direct conclusion may be drawn just form T1-values alone.
Action: Limitations.
“Secondly, T1-tiems reflect a composite alterations within the tissue and T1-mapping results do not allow for direct conclusion about pathophysiological processes. Of note, in NAFLD/MAFLD both fat accumulation and fibrosis play a crucial role, however, fat lowers native hepatic T1-times whereas fibrosis leads to an increase in T1-times. This may explain the weak correlation between CT and CMR tissue characterization findings. Hence, changes in T1-time in the liver may be a result of several components, which could only be confirmed by invasive biopsy..”
R2-3 Please elaborate how T1 timing could differentiate fibrosis from fatty liver. Is magnetic resonance spectroscopy (MRS) relevant in management of MAFLD/NAFLD?
Answer: As fat infiltration leads to significant lowering of overall T1-times within a region of interest, low hepatic T1-times would – theoretically – indicate fatty liver disease. In contrast, fibrosis leads to high T1-times. However, as several pathologies may co-exist (including edema, which also leads to high T1-times), we cannot distinguish between the individual components that result in the mean T1-time of liver tissue.
Action: We expanded the Limitation section (see above) to address this important aspect.
R2-4 Abstract must be improved.
Background: The main hypothesis / purpose of the research are not clear. T1 timing for hepatic tissue characterization has never been performed, but what is the potential benefits of the research? Perhaps the author think that T1 timing could avoid the ionizing radiation of CT, or perhaps the T1 timing could detect fatty liver and cardiac disease in a single scan?
Method: The first sentence in Result "We identified 271 patients ... within 30 days." is part of Method.
Answer and Action: According to our senior statistician, the identification of 271 individuals is already a result given the nature of the study. In the Methods section, the process of patient screening is described. If the reviewer insists, we can move it to the method section.
Discussion: It is not clear if results support the hypothesis of the study?
Answer: Given the explorative character of the study due to lack of pre-existing data, the hypothesis of the study was to test the association between liver characterization on CT versus CMR.
Action: We made respective changes:
Backgound: “We hypothesized that there is a significant correlation between hepatic tissue T1-times on CMR and Hounsfield Units (HU) on non-contrast CT. “
Disucssion: “We report three main findings in this exploratory study of patients undergoing non-contrast CT and CMR within 30 days: 1) liver HU values on CT and native hepatic T1-times on CMR only correlate weakly with each other, 2) liver HU values on CT are associated with cardiometabolic traits, such as diabetes and rather NFS, and 3) in contrast, hepatic T1-times on CMR are associated with impaired cardiac function, including LVEF, RVEF, and natriuretic peptide levels.”
R2-5 Please use consistent terms. It is not clear whether the liver T1 times, the hepatic T1 times, T1 maps and T1-mapping are the same?
Use the same term thoroughout the paper.
Answer: We greatly appreciate this suggestion.
Action: We changed all the “liver t1-times/mapping” into “hepatic”.
R2-6 Background information in Discussion can be moved to Introduction.
Answer and Action: We made respective changes.
R2-7 The main conclusion must be highlighted in accordance with the main hypothesis
Answer: We made respective changes.
Action:
Background: “We hypothesized that there is a significant correlation between hepatic tissue T1-times on CMR and Hounsfield Units (HU) on non-contrast CT. ”
Discussion: “We report three main findings in this exploratory study of patients undergoing non-contrast CT and CMR within 30 days: 1) liver HU values on CT and native hepatic T1-times on CMR only correlate weakly with each other, 2) liver HU values on CT are associated with cardiometabolic traits, such as diabetes and the NFS, and 3) in contrast, hepatic T1-times on CMR are associated with impaired cardiac function, including LVEF, RVEF, and natriuretic peptide levels.”
R2-8 Clearly define Metabolic dysfunction-associated fatty liver disease (MAFLD) and non-alcoholic fatty liver disease (NAFLD) in Introduction.
Answer: We greatly appreciated this important comment. Indeed, the diagnosis of MAFLD/NAFLD is challenging and several methods are currently used. We highlighted this in the revised version of the manuscript.
Action:
“Despite its high prevalence and clinical importance, the diagnosis of MAFLD/NAFLD is challenging. Liver biopsy represents the gold standard for MAFLD/NAFLD diagnosis, but is infrequently performed due to its invasive nature. (…) An NFS of greater than 0.675 indicates severe fibrosis or cirrhosis and a Fib-4 Score greater than or equal to 2.67 indicated advanced fibrosis.(…) CT is commonly used to estimate the risk for MAFLD/NAFD. The attenuation value of liver parenchyma measured in Hounsfield units (HU) is decreased by the presence of hepatic steatosis (…) attenuation values below 40 HU, or a liver-to-spleen HU ratio of <1.0, are often used to define MAFLD/NAFLD. (7)”
R2-9 There are many typos in the article.
Answer and Action: Respective changes were made throughout the manuscript.
R2-10 REFERENCE LIST is Reference
Answer and Action: We changed Reference List to References.
R2-11 Use MDPI template. See https://www.mdpi.com/journal/jcm/instructions
Answer and Action: We revised the entire manuscript to fit the MDPI template and highlight all relevant changes.
Reviewer 3 Report
Overall the manuscript is well written, addressing an interesting topic in a field in which non-invasive diagnosis and prediction of cardiovascular risk is of great importance. I have only a few suggestions:
- Please correct the references' appearance in the whole text, as the number in brackets should appear BEFORE the fullstop of every sentence, not after.
- The p in p-values should be written in lowercase italics; please correct this throughout the text.
- Introduction: "or a ratio of liver-to-spleen HU ratio <1.0" please remove the first "ratio".
- Introduction: "inflammation, oxidative stress, and system insulin resistance". Please correct "system" with "systemic".
- Introduction: the Authors state "So far, only few studies report on the use of T1-mapping in liver MR scans but show promising results. (12)" but then insert only one reference; please add all the correct references for these few studies reporting on the use of T1-mapping in liver MRI.
- Methods: "(see Figure 1)." Please remove "see".
- Methods: "[0,15 mmol/kg gadobutrol". Please change "," with "."
- Methods: please insert the meaning of the acronym "MOLLI".
- Methods: please insert the meaning of the acronym "MDCT".
- Methods: please explain how did you choose the slices in which put the ROIs (how far from Glisson's capsule? how far for main portal branches or hepatic veins?), how large the ROIs were (area or diameter)?).
- Results: please add spaces between numbers and symbols in "55±11"; please correct this throughout the text.
- Results: please add spaces between numbers, symbols and letters in "590±96ms"; please correct this throughout the text.
- Results: "a higher NFS (p=0-008)"; please correct the p-value with "p=0.008".
- The acronyms "gGT" and "GGT" are both present in the manuscript; please choose one and stick to it.
- Results: please insert the meaning of the acronyms "AST, GOT, gGT, NT-proBNP, LVEF, RV, RVEF, RVEDV, BSA, LA, BMI".
- Figure 3 can be removed, as do not add useful informations; in addition, the sentence "Figure 3 summarizes key findings of the study." can also be removed.
- Discussion: "Lee et al. (23) demonstrated that MAFLD/NAFLD was associated with non-calcified coronary plaques and sudden and unexpected cardiac events. (23)". Please remove one of the two references in brackets.
- Discussion: "Bot CT and CMR"; please correct with "both".
Author Response
Point by Point Response
Reviewer 3:
Overall the manuscript is well written, addressing an interesting topic in a field in which non-invasive diagnosis and prediction of cardiovascular risk is of great importance. I have only a few suggestions:
R3-1 Please correct the references' appearance in the whole text, as the number in brackets should appear BEFORE the fullstop of every sentence, not after.
Answer and Action: We made changes throughout the manuscript.
R3-2 The p in p-values should be written in lowercase italics; please correct this throughout the text.
Answer and Action: We changed this accordingly.
R3-3 Introduction: "or a ratio of liver-to-spleen HU ratio <1.0" please remove the first "ratio".
Answer and Action: We changed this accordingly.
R3-4 Introduction: "inflammation, oxidative stress, and system insulin resistance". Please correct "system" with "systemic".
Answer and Action: We changed this accordingly.
R3-5 the Authors state "So far, only few studies report on the use of T1-mapping in liver MR scans but show promising results. (12)" but then insert only one reference; please add all the correct references for these few studies reporting on the use of T1-mapping in liver MRI.
Answer and Action: We changed this accordingly.
R3-6 Methods: "(see Figure 1)." Please remove "see".
Answer and Action: We changed this accordingly.
R3-7 Methods: "[0,15 mmol/kg gadobutrol". Please change "," with "."
Answer and Action: We changed this accordingly.
R3-8 Methods: please insert the meaning of the acronym "MOLLI".
Action: we added: Modified Look-Locker Inversion Recovery (MOLLI)
R3-9 Methods: please insert the meaning of the acronym "MDCT".
Action: we added: Multi-Detector Computed Tomography (MDCT)
R3-10 Methods: please explain how did you choose the slices in which put the ROIs (how far from Glisson's capsule? how far for main portal branches or hepatic veins?), how large the ROIs were (area or diameter)?).
Answer: Thank you for this excellent point. We explained this in more detail.
Action: We added the following sentence in the Methods section: “The regions of interest had a maximum diameter of approximately 1 cm and regions close to vessels or the bile duct were avoided. The distance to the Glisson’s capsule was a minimum of 2 cm.”
R3-11 Results: please add spaces between numbers and symbols in "55±11"; please correct this throughout the text.
Action: done
R3-12 Results: "a higher NFS (p=0-008)"; please correct the p-value with "p=0.008".
Action: done
R3-13 The acronyms "gGT" and "GGT" are both present in the manuscript; please choose one and stick to it.
Answer and Action: We changed this accordingly.
R3-14 Results: please insert the meaning of the acronyms "AST, GOT, gGT, NT-proBNP, LVEF, RV, RVEF, RVEDV, BSA, LA, BMI".
Answer and Action: We changed this accordingly.
R3-15 Figure 3 can be removed, as do not add useful informations; in addition, the sentence "Figure 3 summarizes key findings of the study." can also be removed.
Action: we removed figure 3.
R3-16 Discussion: "Lee et al. (23) demonstrated that MAFLD/NAFLD was associated with non-calcified coronary plaques and sudden and unexpected cardiac events. (23)". Please remove one of the two references in brackets.
Answer and Action: We changed this accordingly.
R3-17 Discussion: "Bot CT and CMR"; please correct with "both".
Answer and Action: We changed this accordingly.
Round 2
Reviewer 2 Report
The revision has been greatly improved. The discussion captures the main findings in this study, which is excellent.
However, there remains many language issues in the paper. Please consider to revise the paper carefully.
Please indent the paragraphes.
Major points:
Line 307: Author contributions:
This section was not edited. Please list the author contributions in detail.
Minor points:
1. I appreciate the English ability of the author. However, multiple minor language errors could diminish the credibility of this paper. Please consider to edit the text. Here are some suggestions:
Line 24: "Mean hepatic HU were 54±11 " could be "Mean hepatic HU values were 54±11 " Mean hepatic attenuation value was 54±11 HU" .
Line 293: Tiems should be times
Line 46: "Liver biopsy represents the gold standard for MAFLD/NAFLD diagnosis, but is infrequently performed due to its invasive nature." The extra "," in the complex sentence seems unnecessary.
Line 40:"a continuum of hepatic disease(s)" ?
Line 64: "(a or the) first-line" ?
Line 73: well-described ?
Line 76: hospital-wide? does it needs the hyphen
Line 80, Line 85: short-axis? the hyphen
Line 169: inter-"rater" variability? Line 149 inter-"reader" variability is appropriate. Please use the same term.
Line 235: Missing s behind cause: "commonly discussed as cause(s) for liver damage "
Line 239: Missing a or the before cause "Cardiac backward failure as (a or the) cause for liver damage can easily "
Line 246: Missing a before cause "Besides heart failure as (a) cause for liver damage,"
Line 304: The s behind warrant is unnecessary "The prognostic value and clinical usefulness of hepatic T1-times in CMR
cohorts warrant() further research."
Author Response
Point by Point Response:
We adapted the manuscript according to the specific formatting comments and the reviewer's comments as listed below.
Reviewer 2:
The revision has been greatly improved. The discussion captures the main findings in this study, which is excellent.
R2-1: Author contributions:
This section was not edited. Please list the author contributions in detail.
Answer: We listed the author contributions correctly.
R2-2: Line 24: "Mean hepatic HU were 54±11 " could be "Mean hepatic HU values were 54±11 " Mean hepatic attenuation value was 54±11 HU" .
Answer: We added the word 'values'.
R2-3: Line 293: Tiems should be times
Answer: We changed it to times.
R2-4: Line 46: "Liver biopsy represents the gold standard for MAFLD/NAFLD diagnosis, but is infrequently performed due to its invasive nature." The extra "," in the complex sentence seems unnecessary.
Answer: We deleted the extra ','.
R2-5: Line 40:"a continuum of hepatic disease(s)" ?
Answer: We added an 's' to disease.
R2-6: Line 64: "(a or the) first-line" ?
Answer: We changed that to 'the' first line imaging.
R2-7: Line 73: well-described ?
Answer: We added the hyphen.
R2-8: Line 76: hospital-wide? does it needs the hyphen
Answer: We deleted the hyphen.
R2-9: Line 80, Line 85: short-axis? the hyphen
Answer: We added the hyphen.
R2-10: Line 169: inter-"rater" variability? Line 149 inter-"reader" variability is appropriate. Please use the same term.
Answer: We changed the term inter-reader to inter-rater in the whole manuscript.
R2-11: Line 235: Missing s behind cause: "commonly discussed as cause(s) for liver damage "
Answer: We added the missing 's'.
R2-12: Line 239: Missing a or the before cause
Answer: We added an 'a' to the sentence: Cardiac backward failure as a cause for liver damage can easily …..
R2-13: Line 304: The s behind warrant is unnecessary "The prognostic value and clinical usefulness of hepatic T1-times in CMR
cohorts warrant() further research."
Answer: We deleted the 's' warrant.